# Metaverse Security: Issues, Challenges and a Viable ZTA Model

Ankur Gupta [1], Habib Ullah Khan [2], Shah Nazir [3], Muhammad Shafiq [4] and Mohammad Shabaz [1,*]

1 Model Institute of Engineering & Technology, Jammu 181122, India
2 Department of Accounting & Information Systems, College of Business and Economics, Qatar University, Doha 122104, Qatar
3 Department of Computer Science, University of Swabi, Swabi 94640, Pakistan
4 Cyberspace Institute of Advance Technology, Guangzhou University, Guangzhou 510006, China
* Correspondence: bhatsab4@gmail.com

**Abstract:** The metaverse is touted as an exciting new technology amalgamation facilitating next-level immersive experiences for users. However, initial experiences indicate that a host of privacy, security and control issues will need to be effectively resolved for its vision to be realized. This paper highlights the security issues that will need to be resolved in the metaverse and the underlying enabling technologies/platforms. It also discussed the broader challenges confronting the developers, the service providers and other stakeholders in the metaverse ecosystem which if left unaddressed may hamper its broad adoption and appeal. Finally, some ideas on building a viable Zero-Trust Architecture (ZTA) model for the metaverse are presented.

**Keywords:** metaverse; metaverse security; privacy in metaverse; metaverse security ecosystem





## 1. Introduction

The metaverse [1,2] concept has captured the imagination of tech-giants, corporates, governments, education service providers and the end consumers themselves as it offers truly immersive experiences leading to the development of an entirely new class of applications and business categories. The metaverse and its potential [3] have even prompted Facebook to re-brand itself as "Meta" [4], signaling its intent to build a virtual-reality-based social network, facilitating social interaction and content consumption in 3D virtual worlds and settings. The metaverse, which consists of several universes called verses, is predicted to be the Internet of the future. Recently, this idea has received a lot of discussion; however, not enough attention has been paid to the security concerns of these virtual worlds. The primary tools used to reach the metaverse are virtual reality headsets. To access the metaverse platforms, a user must authenticate their identity, hence the security of this process is crucial [5]. For the metaverse to draw in a wide audience, it must offer engaging experiences. For the optimum user experience, all user actions must be coordinated and active. The token economy can be used to make the system sustainable. A fictional realm according to some, the metaverse is an upgraded kind of virtual reality technology. However, the term "metaverse" is used to refer to the future Internet, which will include verses, or virtual worlds [6]. It is expected that the metaverse shall have a far-reaching impact on the manner businesses interact with end consumers digitally. Many articles extolling the virtues of the metaverse and its applications to various domains have therefore emerged, ranging from social networks to e-commerce to education, training and skilling to arts and entertainment, to tourism and medicine among several others. A recent survey, which highlights the current state of the art and sets the research agenda for the metaverse, is available in [7].

The metaverse is not a singular concept but an amalgamation of a complete technology ecosystem comprising custom hardware devices, proprietary software platforms and frameworks [8]. While early movers can try and build monopolies in the metaverse space,

the vision of the metaverse can only be realized in a multi-stakeholder model with multiple parties and service providers collaborating to provide the value spectrum for the end users. One critical aspect, which shall govern the widespread adoption of the metaverse, shall be security [9], as well as associated concerns, such as trust, privacy [10] and control. All the stakeholders responsible for delivering the metaverse, including device manufacturers, platform designers/developers, third-party apps and service vendors will need to build novel security and control mechanisms for the metaverse from the ground up. Otherwise, the metaverse runs the risk of compounding the security risks prevalent in all the enabling technologies, such as social networks, cloud computing, open networks, augmented reality (AR), virtual reality (VR) and extended reality (XR) [11]. Through the use of virtual reality and augmented reality technology, the metaverse has come to be understood as a simulated, immersive world that promotes online connection. In order to give people a fictitiously endless experience, virtual and physical places are integrated. Users build virtual avatars that they can use to explore sub-metaverses, or alternative worlds built within the metaverse. By producing compelling signals that evoke psychological and affective responses in the virtual environment, the immersion reality produced by the metaverse aids users in tuning out the real world in favor of a synthetic one. The ability to enjoy those things without ever leaving your home is convenient. This is where virtual reality (VR) comes in; until recently, it was impossible to experience hot, humid rainforests in Africa while sitting and reading about them in Canada. Through VR tools, such as VR headsets, a person can experience what he reads in real time. By following the user's motions and projecting a series of 3D films to create the illusion of a virtual environment, VR offers a simulated experience [12]. In its most basic form, virtual reality (VR) is the process of using a computer to create the 3D world that we perceive with our eyes. A VR headset is utilized so that the user may engage with this virtual world. Gloves or additional apparel can be used to make the experience more intense and realistic. The sensors can track the user's motions, and telepresence (the illusion of "being there") is concurrently established. Augmented reality serves as a link between this virtual world and the actual world, while virtual reality gives us access to 3D places [13].

Zero Trust Architecture (ZTA) [14] is a multi-party decentralized security model suited to open collaborative environments. Some of the prominent elements in the ZTA model are:

- The ability to authenticate everything;
- Repeated time/session-based authentication and identity verification;
- Access control with fine-grained privileges;
- Data validation;
- Traffic and logs validation.

The ZTA model is based on the adage "Do not Trust, Verify", and it is finding wide applicability in delivering end-to-end security in multi-party collaborative environments. Its applicability to secure the metaverse needs to be therefore explored.

This paper presents a comprehensive model of metaverse security issues and challenges by considering the entire metaverse ecosystem as a function of enabling, supporting and collaborating technologies. Potential security threats are examined in detail and prospective solutions are suggested.

The main contributions of this paper are:

- A comprehensive review of the nascent literature related to metaverse security is performed and the major issues are consolidated;
- A comprehensive model for metaverse security is formulated as a function of enabling and supporting technologies;
- A potentially viable Zero-Trust Architecture (ZTA) model to address the identified shortcomings is proposed.

The rest of the paper is organized as follows: Section 2 provides an overview of the technology ecosystem enabling the metaverse. Section 3 presents an overview of the

security challenges in the metaverse, while Section 4 outlines the prospective ZTA model to address the identified security challenges. Finally, Section 5 concludes the paper.

## 2. Literature Survey

The metaverse platform has been utilized for many events, including music concerts, entry ceremonies, new-employee training and virtual real estate auctions. According to analysts, eight of the top ten firms in the world by market cap have entered and begun to compete in the metaverse market. The metaverse has become an open-ended battleground. In 2013, Tencent, a significant Internet business in China, purchased a 40 percent share in Epic Games in the US. According to the investigation, Tencent is consistently investing heavily in platform and content core businesses, which are important sectors for the future metaverse, including games, social media and content. The phrase "metaverse" initially appeared in 1992 [15]. It is a composite word made up of the words meta, which means virtual and transcendental, and universe, which means both world and universe. It is being described as a world where social, economic and cultural activities constantly interact, co-evolve and produce new values. As a result, with the aid of powerful technology from the third and fourth industrial revolutions, the convergence of virtual and reality is progressively becoming a reality. The first and second industrial revolutions brought about innovation in the physical world, whereas the third and fourth industrial revolutions are broadening the area of innovation by integrating the virtual world.

Despite the market's relevance and interest, it is challenging to locate studies that examine the metaverse ecosystem's composition. The idea of the metaverse is still developing, and various individuals are adding to its significance in unique ways [16]. As a result, there are numerous definitions of the term "metaverse" in earlier studies, and there are currently no established guidelines. It was described as a virtual world by some researchers [17], while others characterized it as the fusion of virtual and actual worlds. Additionally, it appears that not many articles have accurately researched the ecosystem of the metaverse. Even though J. Y. Kim [18] studied metaverse ecology, his research was only focused on the content ecosystem. Research is therefore restricted to a certain region of the metaverse. The truth is that the metaverse may be difficult to grasp and apply practically if there is not an established standard for the ecosystem's makeup. The metaverse was tackled and studied from a device perspective in certain studies. There have been additional studies performed from the viewpoint of games or content; however, it is challenging to locate one that analyzes metaverse ecology. The requirement for an approach from the viewpoint of metaverse ecology was highlighted by W. H. Seok. A compound phrase formed from the words "meta," which means virtual and transcendental, and "universe," which means world and universe, describes this metaverse or expanded virtual world. ASF characterized the metaverse as "a universe in which virtual and real constantly interact and co-evolve, in which society and economics form and cultural activities create value" fifteen years later in 2007. Despite receiving a lot of interest, Lind Lab in the US launched the Second Life metaverse service in 2003. Additionally, when Roblox launched its metaverse service in the US in 2006, 55 percent of American youths under the age of 16 signed up, and by 2021, there were 40 million daily users on average—more than three times as many as Facebook, Instagram, YouTube, and TikTok combined [19]. The participant pay system was in the past a weak point in the metaverse economy system. It featured a straightforward revenue model that included social media ads and games. It merely concentrated on transferring the physical environment into a virtual one. In contrast, current and prospective metaverses (such as Roblox's "Robux", Portlight's "V-bucks", ZEPETO's "Coin" and "ZEM" and Decentraland's "MANA") pay and honor players using its own currency [20]. It is a setting where the platform's revenue (through subscription models, content/item sales, revenue from adverts and in-app payment systems) can be used in the actual economy. When creating content, it collaborates with Gucci and Nike to generate B2B revenue. As the four world views are appropriately merged and fused/composited to speed up the interaction, it is usual in practical application to develop a new business model [21]. For

instance, users can wear AR glasses to create a running avatar with the new Ghost Pacer, which was just announced on Kickstarter. The virtual avatar runner sprints in front of the user at a preset speed or at a speed appropriate for his or her previous running. It moves and serves as a pacemaker for the eyes. This is an illustration of how augmented reality (AR) and life journaling have come together. In 2016, Niantic, a Google subsidiary and co-developer of Pokémon GO, conducted research and created related technologies for augmented reality (AR) and virtual reality (VR) [22]. The groundwork has been set to maximize immersion through qualitative advancements in both culture and technology. According to H. S. Ning et al. [23], 96 investigations of the development stage (2020–2021) in the metaverse were conducted from the viewpoints of multiple technologies or ecosystems. The most recent research trend appears to have started looking at the environment of the metaverse. When it comes to the architecture of the metaverse, 2G-, 3G- and 4G-based networks were previously supported, to the point where 2D graphics could be sent via mobile phones and a mouse or keyboard. The current and future metaverse, however, offers a 5G communication network environment, NFTs that can safeguard the ownership of digital assets, and allows GPU-accelerated parallel processing [24]. It is anticipated that Intra will get to the point at which it can effortlessly serve both 2D and 3D content. A potential list of such industries and specific use-cases enabled by the metaverse are included in Table 1 below:

**Table 1.** Metaverse concept and flow of research.

| Research Topic | Main Features | References |
|---|---|---|
| Metaverse Concepts | Snow Crash (1992) proposed the idea of the metaverse, a web-based 3D virtual reality environment where diverse real-world activities are carried out as avatars. | Pınar, K. et al. [5] |
| Metaverse Definition | A virtual environment where users use avatars to participate in social, economic and cultural activities. | Son, K.M. et al. [25] |
| Metaverse Definition | A virtual reality setting where chances for social and economic advancement are provided, similar to actual or virtual worlds | Ryu and Ahn [21] |
| Metaverse Roadmap | Map of the Metaverse using 3D Web was announced and defined as a phenomenon where the real world and virtual world overlap, combine, converge and merge. | Lastowka, G. [26] |
| Metaverse Definition | A setting, technique and entity whereby virtual space and reality engage in active interaction | Suh, S.S. [27] |
| Metaverse Definition | A virtual space world with things, people and connections that live inside a fictitiously determined period. | Schuemie et al. [28] |
| 4th Industrial Revolution | Defined the idea as the convergence of technological revolutions, where the borders between the current digital, physical and biological worlds vanish | Kim, J.Y. [18] |
| Metaverse ecosystem | Stressed the need for metaverse ecosystem studies and characterized the metaverse as a multi-technology confluence | Seok, W.H. [16] |

## 3. The Metaverse Ecosystem

The metaverse encompasses a futuristic vision of virtual worlds delivering highly immersive experiences to consumers while facilitating real-world business to be transacted. It is expected that many industries and business verticals shall be transformed or disrupted by the metaverse. A potential list of such industries and specific use-cases enabled by the metaverse are included in Table 2 below:

**Table 2.** The metaverse and its potential impact on industries.

| Industry | Use Cases/Application Domains |
| --- | --- |
| Sports/Gaming/Fitness | Virtual worlds, digital twins/avatars, immersive viewing and gaming |
| Arts and Entertainment | Virtual tours, 3D OTT and movies, NFTs |
| Decentralized Finance | Virtual trade, seamless B2B and B2C interfaces, real-time financial settlements using blockchain and Decentralized Oracle Networks (DONs) |
| Consumer Electronics | Device evolution, holographs, holograms and 3D projections, headsets, wearables and body implants |
| Communication and Collaboration | Immersive interactions, document management and sharing, business process management, virtual offices |
| Education, Training and Skilling | Immersive simulations, hands-on virtual training, 3D visualizations and remote learning |
| Computing | Personal cloud, GPU pervasiveness in mobile devices and wearables, edge computing and intelligence, AI models for 3D environments, fast 3D rendering and cybersecurity |
| E-Commerce, Fashion and Retail | Virtual stores, immersive product experience, product tryouts, creator-led products, services and economy |
| Industry 4.0 | Remote machinery and plant operations, remote troubleshooting and training |
| Healthcare | Remote robotic procedures, training, simulations, remote diagnosis and immersive tele-medicine |

The metaverse stands to enable several novel business use-case scenarios across a wide variety of industries. Although the metaverse as envisioned does not exist today, several of its enabling supporting technologies do. An amalgamation of these technologies coupled with new developments shall help realize the vision of the metaverse going forward. An early review of the metaverse ecosystem can be found in [29]. Figure 1 presents the overarching technology ecosystem enabling the metaverse. It is abundantly clear that AR/VR/MR and XR technology is the bedrock of the metaverse, allowing users access to a 3D virtual world. In its earliest iteration, the metaverse might be a collection of web 3.0 applications with an XR-Skin providing limited VR experience. Social networks are expected to be among the first to migrate to the metaverse, allowing users to share and consume content immersive along with web 3.0 allowing businesses to offer novel product experiences to users. Blockchain technology is expected to be deployed extensively in the metaverse to enable the vision for decentralized finance and a creator economy, which are emergent themes, primarily due to the security and privacy that they offer [30]. Mobile applications and platforms could be the next to migrate to the metaverse, followed by entertainment platforms (OTT). Finally, 5G/6G connectivity, offering low-latency, could be the glue to tie up all the pieces seamlessly, while the Internet-of-Things connects all devices alongside a heavy use of AI, including edge intelligence [31], to deliver personalized experiences, completing the technology landscape. Of course, APIs or service access points (SAPs) to enable these technologies to collaborate and deliver a complex network of micro-services in the metaverse will still need to be developed.

It is foreseeable that the metaverse ecosystem as shown in Figure 1 may take three potential evolutionary paths:

- Closed Metaverse: For niche applications and use-cases limited to use by a particular community with specialized needs;
- Federated Metaverse: Managed and operated by a large corporation with an ecosystem of cooperating partners, third-party vendors and service providers delivering a unified experience to the end users;
- Open Metaverse: A non-federated metaverse controlled by no single entity with an open architecture and a developer community building applications/services for end users.

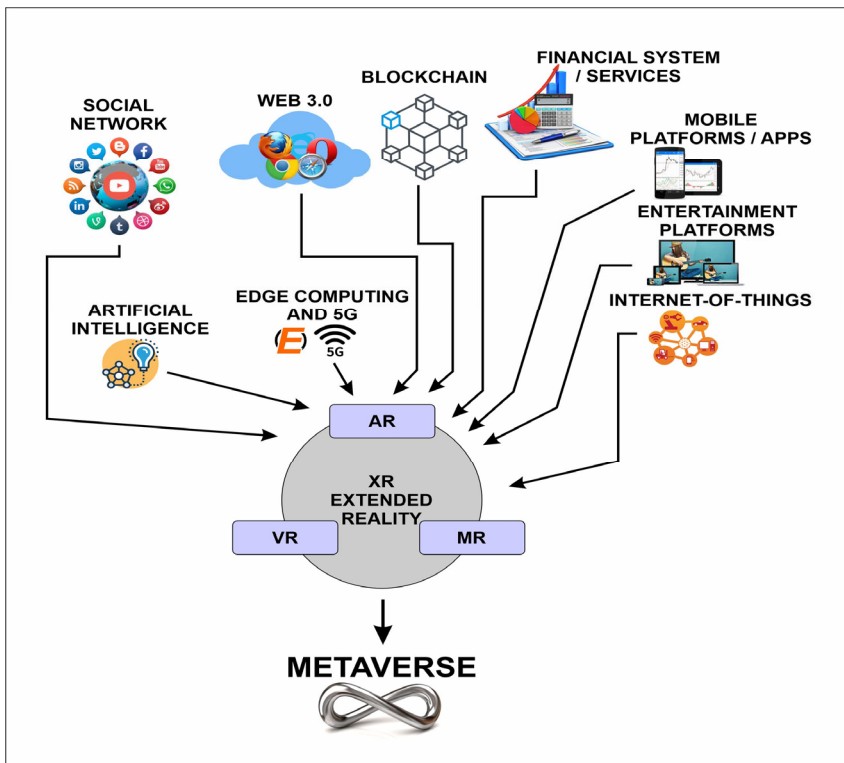

**Figure 1.** The technology ecosystem enabling the metaverse.

All three models can be expected to co-exist in some form or the other. Facebook's Meta is a prime example of the federated metaverse. Other models are expected to emerge in the future, primarily driven by business alliances, mergers and acquisitions.

It is plausible that the metaverse faces several obstacles to its broad adoption. Some of the challenges to the realization of the full potential of the metaverse concept and its adoption include:

- Access: Currently only through headsets, which are not pervasive;
- Ease-of-Use: Users find the current version of headsets bulky and difficult to wear for a long time;
- Lack of Developed Ecosystem: Few Apps available on current VR Platforms;
- Security and Privacy: XR environments suffer from the security vulnerabilities of the underlying technologies, including issues regarding user privacy in virtual worlds.

While advances in technology are expected to address the issues of access and ease of use in the near future, security and privacy issues need to be built from the ground up while designing the metaverse ecosystem. In a recent incident on the Facebook's metaverse prototype, a woman reported being groped and harassed by several male avatars [32]. Such incidents will severely hamper user confidence in adopting the metaverse. If the metaverse adopts the same user anonymity model as existing social networks then issues of online bullying, fake news and hate speech etc., shall be pervasive in the metaverse as well. Hence, it is imperative that user privacy and security should be foundational design elements while designing any metaverse applications, rather than being included as add-ons at a later stage.

Thus, potential security challenges in the metaverse have started to receive the attention of researchers and industry practitioners. Wang et al. [32] reviewed the security and privacy issues in the metaverse and identified all the potential risks from underlying technologies and existing countermeasures. Privacy concerns in the metaverse are also highlighted in [33]. Falchuk et al. [23] highlighted the dangers of tracking user behavior across all connected entities from social networks to smart homes and counters that the metaverse may also permit the same level of user tracking, leading to serious privacy

concerns. Another review on metaverse security [34] focused on user information, communications, scenarios, and goods as four major elements to evaluate security risks against. Some countermeasures were also suggested. The security concerns arising out of the communication framework for the metaverse and how they can be addressed in future 6G networks are highlighted in [35], with several inbuilt security features suggested at the protocol level for metaverse-specific applications. The human aspects of privacy issues in the metaverse are discussed in [36], with a focus on the psychological manipulation and coercion capabilities afforded by a huge amount of personal data available on the metaverse. The authors in [37] examined the adverse impact of the metaverse on human rights, cautioning against its mindless deployment. The security and privacy issues introduced due to 3D modeling and the reconstruction of a human being's avatar are explored in [37], with the authors raising grave concerns around impersonation and misuse/abuse. Similar concerns are raised in [38], wherein the authors explore the security issues around the rapid proliferation of digital twins in the metaverse, citing identity establishment, disambiguation and authentication and its long-term management as the key challenges. The security considerations for the metaverse are critical for nations and governments as well. A case study on the implications of the metaverse on national security is presented in [39], while discourse on having a viable public policy to govern the metaverse can be found in [40]. Thus, metaverse security has become the focus of individuals, national governments and all the technology stakeholders.

It can be seen that while most of the reviews focus on highlighting the security and privacy issues, very few focus on providing solutions apart from summarizing existing security countermeasures in existing technology stacks. A ZTA-model-based security model for the metaverse has not yet been suggested, which is the main contribution of this paper.

## 4. ZTA Model for the Metaverse

Due to the multitude of stakeholders involved in delivering the metaverse experience and the potential vulnerabilities, it is imperative that an open metaverse embrace a Zero-Trust Architecture (ZTA). The ZTA model for the metaverse includes user/entity verification as a core differentiator compared with existing social network models, in order to fix accountability for user actions. Multiple authentication, data validation, audit trails and provenance through the use of blockchain, fine-grained access control, privilege management, continuous monitoring of user/entity behavior, trust and reputation management and finally taking punitive action against erring users/entities form the crux of the proposed ZTA model. These checks and balances are distributed across four layers.

The four-layer model as shown in Figure 2 is explained in detail below:

1. Level 0: Verification and Registration: One of the major challenges in controlling errant user behavior on social networks is the anonymity offered by these platforms. The use of bots is also prevalent in propagating misinformation and fake news. Hence, we propose to make user/entity verification a pre-requisite for the ZTA model for the metaverse. Thus, only pre-verified users/entities shall be permitted to register for the metaverse, thereby alleviating the challenge associated with existing platforms. Twitter had recently introduced the "blue tick" symbol on select user profiles as an indication that the user identity is verified and authentic. This ensured that genuine content creators could obtain credit for their content and imposters would not gain undue benefits. Therefore, there is a case to ensure that all users and entities should have their identity and antecedents verified before being permitted to use public infrastructure and services on the metaverse [41]. While the Twitter process of identity verification is manual, this task can be easily automated. In many countries, governments have built identity verification services over their citizen registries or other such databases, which can be accessed by third-party service providers for a fee [42]. In India, it is common for financial service providers to complete their KYC (Know-Your-Customer) formalities using the Aadhaar (unique ID) identity verification

APIs. It can be expected that verifying user identities and allowing verified users to register on the metaverse shall prevent a large number of untoward incidents common to social networking platforms.

2. Level 1: Identity Management and Authentication: This level performs the first level of authentication before permitting users access to the metaverse. In addition to performing authentication, this layer is also responsible for identity management. In the metaverse, users may want their identities to be publicly visible or they might want to use an avatar or digital twin to access various services. Hence, the identity management function is responsible for creating an avatar or digital twin for the user and storing its mapping in an encrypted database. Hence, user privacy is ensured. Another alternative to delivering trusted identity management in the metaverse is the use of a trusted third-party, which can map the real-identity of the verified user with her digital twin, such that the platform itself is not aware of the actual mapping [43]. Should the need arise, the real-identity can be deciphered using a multi-party request and consensus protocol involving the user, platform, other entities and the trusted third-party identity management service.

3. Level 2: Access Management: The access management layer is responsible for ensuring seamless access to the AR/VR/XR-rendering layer and to the metaverse. The access control layer encompasses the access policies for all the entities in the metaverse to provide fine-grained access control at the level of each entity. Corresponding to the access policies for each entity are the privilege lists, which define increasing privilege levels based on the earned reputation by that entity [44]. The increasing privilege levels seek to reward good user behavior within the metaverse. The access control policies and the privilege lists feed into the level-3 metaverse sentinels, which continuously evaluate user/entity actions and take appropriate actions as per the pre-defined consent and access/privilege levels. At the platform level, access control lists are maintained to control user/entity access to specific areas within the metaverse.

4. Level 3: Control, Trust and Reputation Management: The access and privilege policies, rules and regulations are codified into the rules and regulations engine, which instantiates multiple policy/rules-enforcer agents. User consent in accessing specific areas/content, as well as allowing other users access to the private space of users, is captured by the consent engine. Within the metaverse, the engagement and transactions between different entities, namely business to business, business to consumer and consumer to consumer, are continuously monitored by metaverse sentinels, which work in conjunction with the policy/rules-enforcer agents to detect any violations or unintended behavior by any of the entities. All violations are recorded on the blockchain, and trust and reputation scores are adjusted accordingly. Repeated violations may result in access privileges being revoked or demoted to lower levels. Reputation scores shall be determined for entities over a significant period of time and good scores shall unlock several higher access privileges for users/entities [45,46]. Further, federated machine learning is envisaged to be employed to determine prospective best-fit partners, violators, etc. In this layer for ensuring data validation, the use of Decentralized Oracle Networks (DONs) is also proposed, as DONs ensure credible and authentic off-chain real-world information, which can be used seamlessly with on-chain data within the metaverse. This shall enable exciting decentralized applications (dApps) to be built and deployed while ensuring data quality and security.

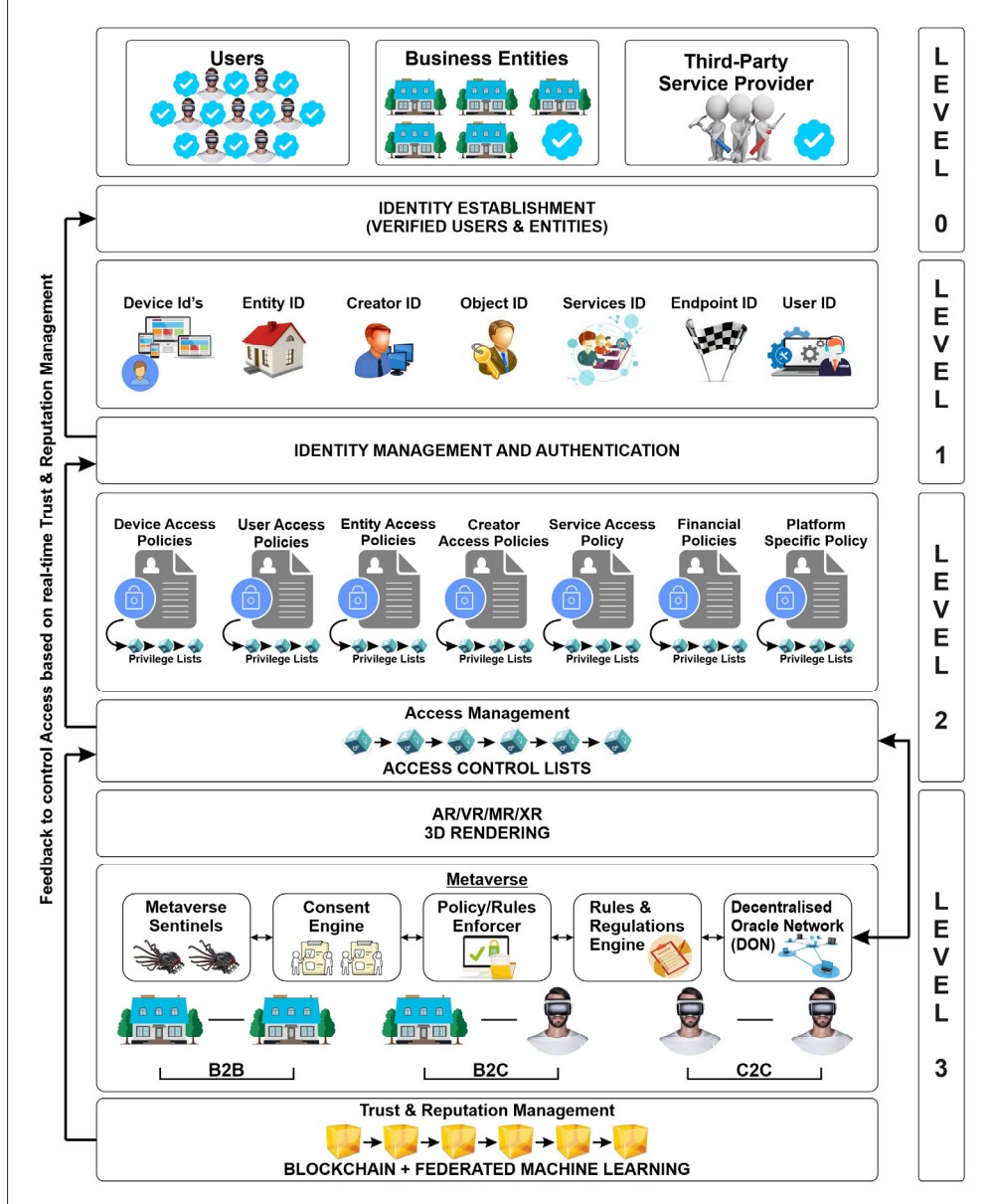

**Figure 2.** A Zero-Trust Architecture model for the metaverse.

Thus, the major elements of the proposed ZTA model for the metaverse are:

- Verified and authenticated identities of users and other entities, promoting greater accountability and provenance;
- Using trusted third-party identity management services, which generate virtual identities for the metaverse while keeping real identities anonymous for enhanced user privacy [47];
- Providing users complete control over their privacy settings and how others can interact with them in the metaverse, including access control to their defined private space;
- Pre-defined rules and regulations outlining acceptable behavior in the metaverse, with the ability to enforce the rules;
- Use of Decentralized Oracle Networks (DONs) as an effective data validation framework for enabling credible off-chain data to be used as part of on-chain data within the metaverse [27];

- Continuous monitoring of user interactions in the metaverse using sentinels and identifying and isolating malicious or undesirable behavior that violates the rules and regulations;
- Computing and managing trust and reputation for all entities in the metaverse on an ongoing basis to govern access rights and privileges [48];
- Ability to remove users from the metaverse based on the severity of the violation committed, leading to permanent revocation of their access [49].

## 5. Conclusions and Future Work

While the metaverse is touted as the evolution of web 3.0, its broad-based adoption shall squarely depend upon whether the platform companies can address issues of security, privacy and trust effectively. Early prototypes have exposed several vulnerabilities, which have cast a shadow on the hype surrounding the future of the metaverse. Any security lapse, identity theft or denial of service in the metaverse can have an impact on the actual world, damaging reputations and hurting commerce. On the other hand, a favorable metaverse experience might improve the organization's real-world operations. The security and privacy precautions being used by the platform provider and the property owner must be understood by consumers. To truly realize the ideal vision of the metaverse, end users would require fool-proof assurances that their privacy shall be protected and that they shall have complete control over how they interact with the metaverse platform and services and, conversely, how other users/entities can interact with them. Allowing, only verified users to register on the platform, providing end users fine-grained control over their experiences within the metaverse, and adopting a ZTA model would alleviate such concerns. The proposed ZTA model is comprehensive and caters for user verification, repeated authentication, identity and access management, control, trust and reputation management, in addition to offering data validation through the use of DONs. Utilizing reputable identity management services from third parties that create metaverse-specific virtual identities while maintaining the anonymity of real identities to improve user privacy in order to allow reliable off-chain data to be used as part of the on-chain data within the metaverse, DONs are used as an efficient data validation mechanism. They have the ability to permanently revoke a user's access and ban them from the metaverse depending on the gravity of the offence they committed. We believe that such a model shall greatly add to the value-perception of the metaverse and help in broadening its adoption base. In future, we will discuss the privacy and security issues surrounding the metaverse. The primary tools used to reach the metaverse are virtual reality headsets. To access the metaverse platforms, a user must authenticate their identity; hence, the security of this process is crucial. Decentralized methods based on blockchain and smart contracts can be used to improve data integrity and guarantee that artificial technology is being applied correctly.

## 6. Future Directions for the Challenges

One must accept that if the appropriate safeguards required for a safe user experience are not built, the hazards that exist within the metaverse could seep into the real world. The potential for using the Internet for illegal purposes, which could cause mental harm rather than physical harm, has an impact on personal safety as well. Online abuse and harassment reports have brought attention to how victims of these crimes can suffer serious psychological effects. In the metaverse, cybercrimes, including cyber-stalking, cyber-bullying and other types of cybercrime, also tend to increase. Inaccuracies in information, fake news and divisive viewpoints may all quickly spread online. One solution to this conundrum is the maintenance of "law and order" within the metaverse and a system of checks and balances for regulators via punishment and reward systems. In order to accomplish the same, an ethical design is required. In addition, the severity and scope of crimes that are committed in the real world and those that are committed in the virtual world have given rise to legal controversy. Building very precise AI systems, such as face and speech recognition that can identify sophisticated physical infiltration and impersonation strategies using deep

fakes and voice simulation, is necessary. A metaverse subsystem's security protocols and firewalls must be built to anticipate cyber-attacks, such as Sybil and DDoS attacks [25,26]. This can be achieved by incorporating conventional antivirus programs into the metaverse or by utilizing cutting-edge machine learning algorithms that not only identify and stop dangerous assaults but also forecast them using metaverse data trends after examining user-generated content (UGC). In virtual reality, virtual network functions can be created and managed by an SDN.

*Research Limitations*

Cyber–physical attacks are one of the major risks to the metaverse, among all other security issues. An attacker can use cyber-attacks to seize control of physical systems. This can entail seizing control of an individual's avatar in order to hurt or change the surroundings. Finding a jurisdiction and a set of laws that can guarantee the users of the virtual environment are safe and secure will be a real problem. The majority of people globally do not have access to broadband internet, which prevents them from taking advantage of the metaverse's full potential [28]. The drawbacks of the metaverse also highlight the need for cutting-edge technology for communication. A premium VR headset, which is needed to access the metaverse, is often out of reach for many individuals.

**Author Contributions:** Conceptualization, A.G.; methodology, A.G.; software, S.N.; validation, A.G., M.S. (Muhammad Shafiq) and H.U.K.; formal analysis, M.S. (Mohammad Shabaz); investigation, A.G.; resources, A.G.; data curation, S.N.; writing—original draft preparation, A.G.; writing—review and editing, M.S. (Mohammad Shabaz); visualization, M.S. (Mohammad Shabaz); supervision, H.U.K.; project administration, M.S. (Mohammad Shabaz); funding acquisition, H.U.K. All authors have read and agreed to the published version of the manuscript.

**Funding:** This research received no external funding.

**Data Availability Statement:** The data shall be made available on request to corresponding author.

**Conflicts of Interest:** The authors declare no conflict of interest.

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
