# Peer review of "Metaverse Security: Issues, Challenges and a Viable ZTA Model"

_electronics, doi:10.3390/electronics12020391_

Round 1

Reviewer 1 Report

Authors have to address  the following suggestions in the review process:

·         The introduction section also lacks sufficient citations. The authors suggest using at least five sources and citing them when discussing topics that go beyond the scope of this Paper.

·         In Section 2:  “The Metaverse Ecosystem”. A potential list of such industries and specific use-cases enabled by the Metaverse is included in Table I; which lack of citations;  There are many websites or use-case links are available to discuss/write about the table content, but the authors have not cited them.

·         There is no literature survey section in the manuscript; is there is a specific reason? For any paper, survey is important to coin the challenges in current technologies.

·         Add a Comparison table at the end of the survey section  and compared with at least 10 to 15

Techniques with appropriate parameters.

·         The implications of the work beyond the scope must be stated in the conclusion section.

·         The future works must be expanded in the conclusion section.

·         Please make sure your paper has the necessary language proofreading.

·         Add a separate section in the manuscript and describe future directions for challenges

·         What are the challenges mentioned and what authors have to provide future directions for the challenges; 

Author Response

Metaverse Security: Issues, Challenges and a Viable ZTA Model

Journal:  Electronics (ISSN 2079-9292)

Manuscript ID: electronics-2047688

Response Letter

Reviewer 1

Comment 1: The introduction section also lacks sufficient citations. The authors suggest using at least five sources and citing them when discussing topics that go beyond the scope of this Paper.

Reply: Authors thanks the esteemed reviewer for the valuable comment. The introduction section has been improved and the suitable citation has been incorporated in the revised manuscript.

The metaverse, which consists of several universes called verses, is predicted to be the Internet of the future. Recently, this idea has received a lot of discussion, but not enough attention has been paid to the security concerns of these virtual worlds. The primary tools used to reach the Metaverse are virtual reality headsets. To access the metaverse platforms, a user must authenticate their identity, hence the security of this process is crucial [19]. For the metaverse to draw in a wide audience, it must offer engaging experiences. For the optimum user experience, all user actions must be coordinated and active. The token economy can be used to make the system sustainable. A fictional realm, according to some, the Metaverse is an upgraded kind of virtual reality technology. However, the term "metaverse" is used to refer to the future Internet, which will include verses, or virtual worlds [37]. Through the use of Virtual Reality and Augmented Reality technology, the Metaverse has come to be understood as a simulated, immersive world that promotes online connection. In order to give people a fictitiously endless experience, virtual and physical places are integrated. Users build virtual avatars that they can use to explore sub-metaverses, or alternative worlds built within the Metaverse. By producing compelling signals that evoke psychological and affective responses in the virtual environment, the immersion reality produced by the Metaverse aids users in tuning out the real world in favour of a synthetic one [25]. The ability to enjoy those things without ever leaving your home is superior. This is where virtual reality (VR) comes in; until recently, it was impossible to read about hot, humid rainforests in Africa while sitting in Canada. Through VR tools like VR headsets, a person can experience what he reads in real-time. By following the user’s motions and projecting a series of 3D films to create the illusion of a virtual environment, VR offers a simulated experience [40].In its most basic form, virtual reality (VR) is the process of using a computer to create the 3D world that we perceive with our eyes. A VR headset is utilised so that the user may engage with this virtual world. Gloves or additional apparel can be used to make the experience more intense and realistic. The sensors can track the user's motions, and telepresence (the illusion of "being there") is concurrently established. Augmented reality serves as a link between this virtual world and the actual world, while virtual reality gives us access to 3D places [1].

[19]. Kürtünlüoğlu, P., Akdik, B., & Karaarslan, E. (2022). Security of virtual reality authentication methods in metaverse: An overview. arXiv preprint arXiv:2209.06447.

[37]. Voshmgir, S. (2020). Token Economy: How the Web3 reinvents the Internet (Vol. 2). Token Kitchen, Access date: 3/11/2021, https://github.com/Token-Economy-Book/

[25] S. Mystakidis, “Metaverse,” Encyclopedia, vol. 2, no. 1, pp. 486–497, 2022

[40] J. M. Zheng, K. W. Chan and I. Gibson, ”Virtual reality,” in IEEE Potentials, vol. 17, no. 2, pp. 20-23, April-May 1998, doi: 10.1109/45.666641

[1] Isakov Abduvokhid Abduvahobovich. Digital educational materials for the organization and conduct of fine arts lessons in uzbekistan. European Multidisciplinary Journal of Modern Science, 4:122–132, 2022

Comment 2:    In Section 2:  “The Metaverse Ecosystem”. A potential list of such industries and specific use-cases enabled by the Metaverse is included in Table I; which lack of citations; There are many websites or use-case links are available to discuss/write about the table content, but the authors have not cited them.

Reply: Authors thanks the esteemed reviewer for suggestion. The following citations has been mentioned with Table 1 in the revised manuscript.

Chung, K. H. Y., Li, D., & Adriaens, P. (2023). Technology-enabled financing of sustainable infrastructure: A case for blockchains and decentralized oracle networks. Technological Forecasting and Social Change187, 122258.

Ott, K. (2009, March). Virtual Reality and Simulations in Adult and Career Education. In Society for Information Technology & Teacher Education International Conference (pp. 1515-1517). Association for the Advancement of Computing in Education (AACE).

Cannavo, A., & Lamberti, F. (2020). How blockchain, virtual reality, and augmented reality are converging, and why. IEEE Consumer Electronics Magazine10(5), 6-13.

Williams, K. (2021). The readiness and compatibility of a modern anode handling and cleaning system for industry 4.0 technologies. In Light Metals 2021 (pp. 957-964). Springer, Cham.

Comment 3: There is no literature survey section in the manuscript; is there is a specific reason? For any paper, survey is important to coin the challenges in current technologies.

Reply: Authors thanks the esteemed reviewer for advice. A new section “Literature survey” has been added in the revised manuscript.

Literature survey

The Metaverse platform has been utilised for many events, including music concerts, entry ceremonies, new employee training, and virtual real estate auctions. Accord- ing to analysts, eight of the top ten firms in the world by market cap have entered and begun to compete in the metaverse market. The metaverse has become an open- ended battleground. In 2013 Tencent, a significant Internet business in China, purchased a 40 percent share in Epic Games in the US. According to the investigation, Tencent is consistently investing heavily in platform and content core businesses, which are important sectors for the future metaverse, including games, social media, and content. The phrase ”metaverse” initially appeared in 1992’s Snow Crash by Neil Stevenson [8]. It is a composite word made up of the words meta, which means virtual, transcendental, and universe, which means both world and universe. It is being described as a world where social, economic, and cultural activities constantly interact, co-evolve, and produce new values. As a result, with the aid of powerful technology from the third and fourth industrial revolutions, the convergence of virtual and reality is progressively be- coming a reality. The first and second industrial revolutions brought about innovation in the physical world, whereas the third and fourth industrial revolutions are broadening the area of innovation by integrating the virtual world. Despite the market’s relevance and interest, it is challenging to locate studies that examine the metaverse ecosystem’s com- position. The idea of the metaverse is still developing, and various individuals are adding to its significance in unique ways [22]. As a result, there are numerous definitions of the term ”metaverse” in the majority of earlier studies, and there are currently no established guidelines. It was described as a virtual world by some researchers [16], while others characterised it as the fusion of virtual and actual worlds. Additionally, it appears that not many articles have gone a step farther and accurately researched the ecosystem of the metaverse. Even though J. Y. Kim [17] studied the metaverse ecology his research was only focused on the content ecosystem. Research is therefore restricted to a certain region of the metaverse. The truth is that the metaverse may be difficult to grasp and apply practically if there isn’t an established standard for the ecosystem’s makeup. The metaverse was tackled and studied from the device side in certain studies [16]. There have been additional studies done from the viewpoint of games or contents, but it is challenging to locate one that looked at the metaverse ecology as a whole. The requirement for an approach from the viewpoint of the metaverse ecology was highlighted by W. H. Seok in the year [32]. A compound phrase formed from the words ”meta,” which means virtual and transcendental, and ”universe,” which means world and universe, describes this metaverse or expanded virtual world [13]. ASF characterised the metaverse as ”a universe in which virtual and real constantly interact and co-evolve, in which society and economics form and cultural activities create value” fifteen years later, in 2007. Despite receiving a lot of interest, Lind Lab in the US launched the Second Life metaverse service in 2003. Additionally, when Roblox launched its metaverse service in the US in 2006, 55 percent of American youths under the age of 16 signed up, and by 2021, there were 40 million daily users on average. More than three times as many as Facebook, Instagram, YouTube, and TikTok combined. It turns out to take quite a bit of time [7]. The participant pay system was in the past a weak point in the metaverse economy system. It featured a straightforward revenue model that included social media ads and games. It merely concentrated on transferring the physical environment into a virtual one. In contrast, current and prospective metaverses (such as Roblox’s ”Robux,” Portlight’s ”V-bucks,” ZEPETO’s ”Coin” and ”ZEM,” and Decentraland’s ”MANA”) pay and honour players using its own currency [18]. It is a setting where the platform’s revenue (through subscription models, content/item sales, revenue from adverts, and in-app payment systems) can be used in the actual economy. When creating content, it collaborates with Gucci and Nike to generate B2B revenue. As the four world views are appropriately merged and fused/composited to speed up the interaction, it is usual in practical application to develop a new business model [31]. For instance, Users can wear AR glasses to create a running avatar with the new Ghost Pacer, which was just announced on Kickstarter [23]. The virtual avatar runner sprints in front of the user at a preset speed or at a speed appropriate for his or her previous running. It moves and serves as a pacemaker for the eyes. This is an illustration of how augmented reality (AR) and life journaling have come together. In 2016, Niantic, a Google subsidiary and co-developer of Pokemon GO, conducted research and created related technologies for augmented reality (AR) and virtual reality (VR) [28]. The groundwork has been set to maximise immersion through qualitative advancements in both culture and technology. According to H. S. Ning et al. [27], 96 investigations of the development stage (2020–2021) in the metaverse were conducted from the viewpoints of multiple technologies or ecosystems. The most recent research trend appears to have started looking at the environment of the metaverse. When it comes to the architecture of the metaverse, 2G, 3G, and 4G-based networks were previously supported, to the point where 2D graphics could be sent via mobile phones and a mouse or keyboard. The current and future metaverse, however, offers a 5G communication network environment an NFT that can safeguard the ownership of digital assets,

Comment 4: Add a Comparison table at the end of the survey section and compared with at least 10 to 15

Reply: Author thanks the reviewer for the comments. Comparison table has been added at the end of literature review section.

Table 1.  Metaverse concept and flow of research.

Research topic

Main Features

References

Metaverse Concepts

Snow Crash (1992), proposed the idea of the metaverse, a web-based 3D virtual reality environment where diverse real-world activities are carried out as avatars.

K. Pınar et al.

Metaverse Definition

A virtual environment where users use avatars to participate in social, economic, and cultural activities.

K.M.  Son  et

al.(2006)

Metaverse Definition

a virtual reality setting where chances for social and economic advancement are provided, much like in actual or virtual worlds

Ryu  &

Ahn

Metaverse Roadmap

Map of the Metaverse the way to the 3D Web was announced defined as a phenomenon where the real world and the virtual world overlap, combine, converge, and merge.

G. Lastowka,

Metaverse Definition

A setting, technique, and entity whereby virtual space and reality engage in active interaction

S.  S.

Suh

Metaverse Definition

A virtual space world with things, people, and connections that live inside a fictitiously determined period.

Schuemie et al.

4th Industrial Revolution

defining the idea as the convergence of technological revolutions when the borders between the current digital, physical, and biological worlds vanish

J. Y. Kim. 

Metaverse ecosystem

stressing the need for metaverse ecosystem study and characterising the metaverse as a multi-technology confluence

W.  H.  Seok.

Kurtunluoglu, Pınar, Beste Akdik, and Enis Karaarslan. ”Security of Virtual Reality Authentication Methods in Metaverse: An Overview.”arXiv preprint arXiv:2209.06447 (2022).

  1. M. Son & B.  R.  Lee & K.  H.  Shim & K.  H.  Yang. (2006).  Matrix  World  Metaverse  Created  by  Web  2.0 and  Online  Games,  ETRI  CEO  Information  No.  47, 1-26
  2. Y. Ryu & J.  K. Ahn. (2007). A Study on Digital Storytelling in the Virtual World, Journal of the Game Industry, No.  1, 30-47
  3. Lastowka, “User-generated content and virtual worlds,” Vand. J. Ent. & Tech. L., vol. 10, p. 893, 2007
  4. S. Suh(2008),  A  Study  on  Metaverse  DevelopmentTrends  and  Prospects,  Korean  HCI  Society, Conference, 1450-1457.
  5. J. Schuemie, P. Van Der Straaten, M. Krijn, and C. A. Van Der Mast,“Research on presence in virtual reality: A survey,” CyberPsychology & Behavior, vol. 4, no. 2, pp. 183–201, 2001
  6. Y. Kim. (2021). Metaverse content innovation ecosystem and conditions for sustainable growth .Science and Technology Policy Institute, Future horizon, 25-30
  7. H. Seok. (2021). Analysis of meterverse business model and ecosystem, ETRI, DOI: https;//doi,ogr/10_22648/ETRI.2021.J.360408

Comment 5: The implications of the work beyond the scope must be stated in the conclusion section.

Reply: Authors acknowledge the reviewer comment. The implications of the work beyond the scope has been incorporated in the conclusion.

Any security lapse, identity theft, or denial of service in the metaverse can have an impact on the actual world, damaging reputations and hurting commerce. On the other hand, a favourable metaverse experience might improve the organization's real-world operations. The security and privacy precautions being used by the platform provider and the property owner must be understood by consumers.

Comment 6: The future works must be expanded in the conclusion section.

Reply: Authors thanks the esteemed reviewer for advice. Conclusion section has been enhanced in the revised manuscript by adding the future scope of the work.

In future we will discuss about the privacy and security issues with the metaverse. The primary tools used to reach the Metaverse are virtual reality headsets. To access the metaverse platforms, a user must authenticate their identity, hence the security of this process is crucial. Decentralized methods based on blockchain and smart contracts can be used to improve data integrity and guarantee that artificial technology is being applied correctly.

Comment 7: Please make sure your paper has the necessary language proofreading.

Reply: Author thanks the reviewer for the comments. The manuscript has been thoroughly revised and checked for typographical and grammatical errors and all such errors have been rectified.

Comment 8: Add a separate section in the manuscript and describe future directions for challenges.

Reply: Authors thanks the esteemed reviewer for advice. A new section “Future directions for the challenges” has been added in the revised manuscript.

Future directions for the challenges

 One solution to this conundrum is the maintenance of "law and order" within the Metaverse and a system of checks and balances for regulators via punishment and reward systems. In order to accomplish the same, an ethical design is required. In addition, the severity and scope of crimes that are committed in the real world and those that are committed in the virtual world have given rise to legal controversy. Building very precise AI systems like face and speech recognition that can identify sophisticated physical infiltration and impersonation strategies like using deep fakes and voice simulation is necessary. A metaverse subsystem's security protocols and firewalls must be built with anticipated cyberattacks like Sybil and DDoS attacks [15] in mind. This can be achieved by incorporating conventional antivirus programmes into the metaverse or by utilising cutting-edge machine learning algorithms that not only identify and stop dangerous assaults but also forecast them using metaverse data trends after examining User Generated Content, or UGC. In virtual reality, virtual network functions can be created and managed by an SDN.

Comment 9: What are the challenges mentioned and what authors have to provide future directions for the challenges; 

Reply: Authors thanks the esteemed reviewer for advice. A new section “Future directions for the challenges” has been added in the revised manuscript.

One must accept that if the appropriate safeguards required for a safe user experience are not built, the hazards that exist within the Metaverse could seep into the real world. The potential for using the internet for illegal purposes that could cause mental harm rather than physical harm has an impact on personal safety as well. Online abuse and harassment reports have brought attention to how victims of these crimes can suffer serious psychological effects. In the Metaverse, cybercrimes including cyberstalking, cyberbullying, and other types of cybercrime also tend to increase. Inaccuracies in information, fake news, and divisive viewpoints may all quickly spread online.

Future directions for the challenges

 One solution to this conundrum is the maintenance of "law and order" within the Metaverse and a system of checks and balances for regulators via punishment and reward systems. In order to accomplish the same, an ethical design is required. In addition, the severity and scope of crimes that are committed in the real world and those that are committed in the virtual world have given rise to legal controversy. Building very precise AI systems like face and speech recognition that can identify sophisticated physical infiltration and impersonation strategies like using deep fakes and voice simulation is necessary. A metaverse subsystem's security protocols and firewalls must be built with anticipated cyberattacks like Sybil and DDoS attacks [15] in mind. This can be achieved by incorporating conventional antivirus programmes into the metaverse or by utilising cutting-edge machine learning algorithms that not only identify and stop dangerous assaults but also forecast them using metaverse data trends after examining User Generated Content, or UGC. In virtual reality, virtual network functions can be created and managed by an SDN.

Reviewer 2 Report

 This manuscript highlighted the security issues that will need to be resolved in the metaverse and the underlying enabling technologies/platforms as well as proposed a Zero-Trust Architecture (ZTA) model for the metaverse. The manuscript iscientifically sound and the authors managed to address reliable relevant and recent papers related to the topic. There is a ZTA framework proposed by the author but there is no implementation nor results so we can’t 100% say that this particular framework can solve any security or privacy concerns associated to Metaverse. 

Author Response

Metaverse Security: Issues, Challenges and a Viable ZTA Model

Journal:  Electronics (ISSN 2079-9292)

Manuscript ID: electronics-2047688

Response Letter

Reviewer 2

This manuscript highlighted the security issues that will need to be resolved in the metaverse and the underlying enabling technologies/platforms as well as proposed a Zero-Trust Architecture (ZTA) model for the metaverse. The manuscript iscientifically sound and the authors managed to address reliable relevant and recent papers related to the topic. There is a ZTA framework proposed by the author but there is no implementation nor results so we can’t 100% say that this particular framework can solve any security or privacy concerns associated to Metaverse. 

Reply: Authors thanks the esteemed reviewer for the valuable comment. The manuscript has been improved with adding the sections such as literature survey, Future directions for the challenges, Table etc. which increase the overall quality of the manuscript.

Comment 1: The introduction section also lacks sufficient citations. The authors suggest using at least five sources and citing them when discussing topics that go beyond the scope of this Paper.

Reply: Authors thanks the esteemed reviewer for the valuable comment. The introduction section has been improved and the suitable citation has been incorporated in the revised manuscript.

The metaverse, which consists of several universes called verses, is predicted to be the Internet of the future. Recently, this idea has received a lot of discussion, but not enough attention has been paid to the security concerns of these virtual worlds. The primary tools used to reach the Metaverse are virtual reality headsets. To access the metaverse platforms, a user must authenticate their identity, hence the security of this process is crucial [19]. For the metaverse to draw in a wide audience, it must offer engaging experiences. For the optimum user experience, all user actions must be coordinated and active. The token economy can be used to make the system sustainable. A fictional realm, according to some, the Metaverse is an upgraded kind of virtual reality technology. However, the term "metaverse" is used to refer to the future Internet, which will include verses, or virtual worlds [37]. Through the use of Virtual Reality and Augmented Reality technology, the Metaverse has come to be understood as a simulated, immersive world that promotes online connection. In order to give people a fictitiously endless experience, virtual and physical places are integrated. Users build virtual avatars that they can use to explore sub-metaverses, or alternative worlds built within the Metaverse. By producing compelling signals that evoke psychological and affective responses in the virtual environment, the immersion reality produced by the Metaverse aids users in tuning out the real world in favour of a synthetic one [25]. The ability to enjoy those things without ever leaving your home is superior. This is where virtual reality (VR) comes in; until recently, it was impossible to read about hot, humid rainforests in Africa while sitting in Canada. Through VR tools like VR headsets, a person can experience what he reads in real-time. By following the user’s motions and projecting a series of 3D films to create the illusion of a virtual environment, VR offers a simulated experience [40].In its most basic form, virtual reality (VR) is the process of using a computer to create the 3D world that we perceive with our eyes. A VR headset is utilised so that the user may engage with this virtual world. Gloves or additional apparel can be used to make the experience more intense and realistic. The sensors can track the user's motions, and telepresence (the illusion of "being there") is concurrently established. Augmented reality serves as a link between this virtual world and the actual world, while virtual reality gives us access to 3D places [1].

[19]. Kürtünlüoğlu, P., Akdik, B., & Karaarslan, E. (2022). Security of virtual reality authentication methods in metaverse: An overview. arXiv preprint arXiv:2209.06447.

[37]. Voshmgir, S. (2020). Token Economy: How the Web3 reinvents the Internet (Vol. 2). Token Kitchen, Access date: 3/11/2021, https://github.com/Token-Economy-Book/

[25] S. Mystakidis, “Metaverse,” Encyclopedia, vol. 2, no. 1, pp. 486–497, 2022

[40] J. M. Zheng, K. W. Chan and I. Gibson, ”Virtual reality,” in IEEE Potentials, vol. 17, no. 2, pp. 20-23, April-May 1998, doi: 10.1109/45.666641

[1] Isakov Abduvokhid Abduvahobovich. Digital educational materials for the organization and conduct of fine arts lessons in uzbekistan. European Multidisciplinary Journal of Modern Science, 4:122–132, 2022

Comment 2:    In Section 2:  “The Metaverse Ecosystem”. A potential list of such industries and specific use-cases enabled by the Metaverse is included in Table I; which lack of citations; There are many websites or use-case links are available to discuss/write about the table content, but the authors have not cited them.

Reply: Authors thanks the esteemed reviewer for suggestion. The following citations has been mentioned with Table 1 in the revised manuscript.

Chung, K. H. Y., Li, D., & Adriaens, P. (2023). Technology-enabled financing of sustainable infrastructure: A case for blockchains and decentralized oracle networks. Technological Forecasting and Social Change187, 122258.

Ott, K. (2009, March). Virtual Reality and Simulations in Adult and Career Education. In Society for Information Technology & Teacher Education International Conference (pp. 1515-1517). Association for the Advancement of Computing in Education (AACE).

Cannavo, A., & Lamberti, F. (2020). How blockchain, virtual reality, and augmented reality are converging, and why. IEEE Consumer Electronics Magazine10(5), 6-13.

Williams, K. (2021). The readiness and compatibility of a modern anode handling and cleaning system for industry 4.0 technologies. In Light Metals 2021 (pp. 957-964). Springer, Cham.

Comment 3: There is no literature survey section in the manuscript; is there is a specific reason? For any paper, survey is important to coin the challenges in current technologies.

Reply: Authors thanks the esteemed reviewer for advice. A new section “Literature survey” has been added in the revised manuscript.

Literature survey

The Metaverse platform has been utilised for many events, including music concerts, entry ceremonies, new employee training, and virtual real estate auctions. Accord- ing to analysts, eight of the top ten firms in the world by market cap have entered and begun to compete in the metaverse market. The metaverse has become an open- ended battleground. In 2013 Tencent, a significant Internet business in China, purchased a 40 percent share in Epic Games in the US. According to the investigation, Tencent is consistently investing heavily in platform and content core businesses, which are important sectors for the future metaverse, including games, social media, and content. The phrase ”metaverse” initially appeared in 1992’s Snow Crash by Neil Stevenson [8]. It is a composite word made up of the words meta, which means virtual, transcendental, and universe, which means both world and universe. It is being described as a world where social, economic, and cultural activities constantly interact, co-evolve, and produce new values. As a result, with the aid of powerful technology from the third and fourth industrial revolutions, the convergence of virtual and reality is progressively be- coming a reality. The first and second industrial revolutions brought about innovation in the physical world, whereas the third and fourth industrial revolutions are broadening the area of innovation by integrating the virtual world. Despite the market’s relevance and interest, it is challenging to locate studies that examine the metaverse ecosystem’s com- position. The idea of the metaverse is still developing, and various individuals are adding to its significance in unique ways [22]. As a result, there are numerous definitions of the term ”metaverse” in the majority of earlier studies, and there are currently no established guidelines. It was described as a virtual world by some researchers [16], while others characterised it as the fusion of virtual and actual worlds. Additionally, it appears that not many articles have gone a step farther and accurately researched the ecosystem of the metaverse. Even though J. Y. Kim [17] studied the metaverse ecology his research was only focused on the content ecosystem. Research is therefore restricted to a certain region of the metaverse. The truth is that the metaverse may be difficult to grasp and apply practically if there isn’t an established standard for the ecosystem’s makeup. The metaverse was tackled and studied from the device side in certain studies [16]. There have been additional studies done from the viewpoint of games or contents, but it is challenging to locate one that looked at the metaverse ecology as a whole. The requirement for an approach from the viewpoint of the metaverse ecology was highlighted by W. H. Seok in the year [32]. A compound phrase formed from the words ”meta,” which means virtual and transcendental, and ”universe,” which means world and universe, describes this metaverse or expanded virtual world [13]. ASF characterised the metaverse as ”a universe in which virtual and real constantly interact and co-evolve, in which society and economics form and cultural activities create value” fifteen years later, in 2007. Despite receiving a lot of interest, Lind Lab in the US launched the Second Life metaverse service in 2003. Additionally, when Roblox launched its metaverse service in the US in 2006, 55 percent of American youths under the age of 16 signed up, and by 2021, there were 40 million daily users on average. More than three times as many as Facebook, Instagram, YouTube, and TikTok combined. It turns out to take quite a bit of time [7]. The participant pay system was in the past a weak point in the metaverse economy system. It featured a straightforward revenue model that included social media ads and games. It merely concentrated on transferring the physical environment into a virtual one. In contrast, current and prospective metaverses (such as Roblox’s ”Robux,” Portlight’s ”V-bucks,” ZEPETO’s ”Coin” and ”ZEM,” and Decentraland’s ”MANA”) pay and honour players using its own currency [18]. It is a setting where the platform’s revenue (through subscription models, content/item sales, revenue from adverts, and in-app payment systems) can be used in the actual economy. When creating content, it collaborates with Gucci and Nike to generate B2B revenue. As the four world views are appropriately merged and fused/composited to speed up the interaction, it is usual in practical application to develop a new business model [31]. For instance, Users can wear AR glasses to create a running avatar with the new Ghost Pacer, which was just announced on Kickstarter [23]. The virtual avatar runner sprints in front of the user at a preset speed or at a speed appropriate for his or her previous running. It moves and serves as a pacemaker for the eyes. This is an illustration of how augmented reality (AR) and life journaling have come together. In 2016, Niantic, a Google subsidiary and co-developer of Pokemon GO, conducted research and created related technologies for augmented reality (AR) and virtual reality (VR) [28]. The groundwork has been set to maximise immersion through qualitative advancements in both culture and technology. According to H. S. Ning et al. [27], 96 investigations of the development stage (2020–2021) in the metaverse were conducted from the viewpoints of multiple technologies or ecosystems. The most recent research trend appears to have started looking at the environment of the metaverse. When it comes to the architecture of the metaverse, 2G, 3G, and 4G-based networks were previously supported, to the point where 2D graphics could be sent via mobile phones and a mouse or keyboard. The current and future metaverse, however, offers a 5G communication network environment an NFT that can safeguard the ownership of digital assets,

Comment 4: Add a Comparison table at the end of the survey section and compared with at least 10 to 15

Reply: Author thanks the reviewer for the comments. Comparison table has been added at the end of literature review section.

Table 1.  Metaverse concept and flow of research.

Research topic

Main Features

References

Metaverse Concepts

Snow Crash (1992), proposed the idea of the metaverse, a web-based 3D virtual reality environment where diverse real-world activities are carried out as avatars.

K. Pınar et al.

Metaverse Definition

A virtual environment where users use avatars to participate in social, economic, and cultural activities.

K.M.  Son  et

al.(2006)

Metaverse Definition

a virtual reality setting where chances for social and economic advancement are provided, much like in actual or virtual worlds

Ryu  &

Ahn

Metaverse Roadmap

Map of the Metaverse the way to the 3D Web was announced defined as a phenomenon where the real world and the virtual world overlap, combine, converge, and merge.

G. Lastowka,

Metaverse Definition

A setting, technique, and entity whereby virtual space and reality engage in active interaction

S.  S.

Suh

Metaverse Definition

A virtual space world with things, people, and connections that live inside a fictitiously determined period.

Schuemie et al.

4th Industrial Revolution

defining the idea as the convergence of technological revolutions when the borders between the current digital, physical, and biological worlds vanish

J. Y. Kim. 

Metaverse ecosystem

stressing the need for metaverse ecosystem study and characterising the metaverse as a multi-technology confluence

W.  H.  Seok.

Kurtunluoglu, Pınar, Beste Akdik, and Enis Karaarslan. ”Security of Virtual Reality Authentication Methods in Metaverse: An Overview.”arXiv preprint arXiv:2209.06447 (2022).

  1. M. Son & B.  R.  Lee & K.  H.  Shim & K.  H.  Yang. (2006).  Matrix  World  Metaverse  Created  by  Web  2.0 and  Online  Games,  ETRI  CEO  Information  No.  47, 1-26
  2. Y. Ryu & J.  K. Ahn. (2007). A Study on Digital Storytelling in the Virtual World, Journal of the Game Industry, No.  1, 30-47
  3. Lastowka, “User-generated content and virtual worlds,” Vand. J. Ent. & Tech. L., vol. 10, p. 893, 2007
  4. S. Suh(2008),  A  Study  on  Metaverse  DevelopmentTrends  and  Prospects,  Korean  HCI  Society, Conference, 1450-1457.
  5. J. Schuemie, P. Van Der Straaten, M. Krijn, and C. A. Van Der Mast,“Research on presence in virtual reality: A survey,” CyberPsychology & Behavior, vol. 4, no. 2, pp. 183–201, 2001
  6. Y. Kim. (2021). Metaverse content innovation ecosystem and conditions for sustainable growth .Science and Technology Policy Institute, Future horizon, 25-30
  7. H. Seok. (2021). Analysis of meterverse business model and ecosystem, ETRI, DOI: https;//doi,ogr/10_22648/ETRI.2021.J.360408

Comment 5: The implications of the work beyond the scope must be stated in the conclusion section.

Reply: Authors acknowledge the reviewer comment. The implications of the work beyond the scope has been incorporated in the conclusion.

Any security lapse, identity theft, or denial of service in the metaverse can have an impact on the actual world, damaging reputations and hurting commerce. On the other hand, a favourable metaverse experience might improve the organization's real-world operations. The security and privacy precautions being used by the platform provider and the property owner must be understood by consumers.

Comment 6: The future works must be expanded in the conclusion section.

Reply: Authors thanks the esteemed reviewer for advice. Conclusion section has been enhanced in the revised manuscript by adding the future scope of the work.

In future we will discuss about the privacy and security issues with the metaverse. The primary tools used to reach the Metaverse are virtual reality headsets. To access the metaverse platforms, a user must authenticate their identity, hence the security of this process is crucial. Decentralized methods based on blockchain and smart contracts can be used to improve data integrity and guarantee that artificial technology is being applied correctly.

Comment 7: Please make sure your paper has the necessary language proofreading.

Reply: Author thanks the reviewer for the comments. The manuscript has been thoroughly revised and checked for typographical and grammatical errors and all such errors have been rectified.

Comment 8: Add a separate section in the manuscript and describe future directions for challenges.

Reply: Authors thanks the esteemed reviewer for advice. A new section “Future directions for the challenges” has been added in the revised manuscript.

Future directions for the challenges

 One solution to this conundrum is the maintenance of "law and order" within the Metaverse and a system of checks and balances for regulators via punishment and reward systems. In order to accomplish the same, an ethical design is required. In addition, the severity and scope of crimes that are committed in the real world and those that are committed in the virtual world have given rise to legal controversy. Building very precise AI systems like face and speech recognition that can identify sophisticated physical infiltration and impersonation strategies like using deep fakes and voice simulation is necessary. A metaverse subsystem's security protocols and firewalls must be built with anticipated cyberattacks like Sybil and DDoS attacks [15] in mind. This can be achieved by incorporating conventional antivirus programmes into the metaverse or by utilising cutting-edge machine learning algorithms that not only identify and stop dangerous assaults but also forecast them using metaverse data trends after examining User Generated Content, or UGC. In virtual reality, virtual network functions can be created and managed by an SDN.

Comment 9: What are the challenges mentioned and what authors have to provide future directions for the challenges; 

Reply: Authors thanks the esteemed reviewer for advice. A new section “Future directions for the challenges” has been added in the revised manuscript.

One must accept that if the appropriate safeguards required for a safe user experience are not built, the hazards that exist within the Metaverse could seep into the real world. The potential for using the internet for illegal purposes that could cause mental harm rather than physical harm has an impact on personal safety as well. Online abuse and harassment reports have brought attention to how victims of these crimes can suffer serious psychological effects. In the Metaverse, cybercrimes including cyberstalking, cyberbullying, and other types of cybercrime also tend to increase. Inaccuracies in information, fake news, and divisive viewpoints may all quickly spread online.

Future directions for the challenges

 One solution to this conundrum is the maintenance of "law and order" within the Metaverse and a system of checks and balances for regulators via punishment and reward systems. In order to accomplish the same, an ethical design is required. In addition, the severity and scope of crimes that are committed in the real world and those that are committed in the virtual world have given rise to legal controversy. Building very precise AI systems like face and speech recognition that can identify sophisticated physical infiltration and impersonation strategies like using deep fakes and voice simulation is necessary. A metaverse subsystem's security protocols and firewalls must be built with anticipated cyberattacks like Sybil and DDoS attacks [15] in mind. This can be achieved by incorporating conventional antivirus programmes into the metaverse or by utilising cutting-edge machine learning algorithms that not only identify and stop dangerous assaults but also forecast them using metaverse data trends after examining User Generated Content, or UGC. In virtual reality, virtual network functions can be created and managed by an SDN.

Reviewer 3 Report

This manuscript studied the security problem in the metaverse and the underlying enabling technologies, and proposed a viable Zero-Trust Architecture (ZTA) model for the metaverse are presented, which facilitates addressing safety and control issues.  However, there is still great space for improvement in your manuscript workload, which must be solved before it is considered for publication.

1、 Relevant metaverse security research background needs to be supplemented in INTRODUCTION, or add chapter "Related work".

2、The authors are suggested to improve the authority of references, and you should cite all papers you use properly.  

3、This manuscript should be added with the part of Research Limitations.

4、CONCLUSIONS needs more in it, as it's more of an afterthought.   The authors are suggested to highlight important findings and include afterthought of this work.

 5、 The authors are suggested to enrich the contents of the whole manuscript including tables, to make it more convincing.

6、Please carefully check some small writing problems. For example, missing a "." in line 42.

Author Response

Metaverse Security: Issues, Challenges and a Viable ZTA Model

Journal:  Electronics (ISSN 2079-9292)

Manuscript ID: electronics-2047688

Response Letter

Reviewer 3

Comment 1: Relevant metaverse security research background needs to be supplemented in INTRODUCTION, or add chapter "Related work".

Reply: Authors thanks the esteemed reviewer for advice. A new section “Literature survey” has been added in the revised manuscript.

Literature survey

The Metaverse platform has been utilised for many events, including music concerts, entry ceremonies, new employee training, and virtual real estate auctions. Accord- ing to analysts, eight of the top ten firms in the world by market cap have entered and begun to compete in the metaverse market. The metaverse has become an open- ended battleground. In 2013 Tencent, a significant Internet business in China, purchased a 40 percent share in Epic Games in the US. According to the investigation, Tencent is consistently investing heavily in platform and content core businesses, which are important sectors for the future metaverse, including games, social media, and content. The phrase ”metaverse” initially appeared in 1992’s Snow Crash by Neil Stevenson [8]. It is a composite word made up of the words meta, which means virtual, transcendental, and universe, which means both world and universe. It is being described as a world where social, economic, and cultural activities constantly interact, co-evolve, and produce new values. As a result, with the aid of powerful technology from the third and fourth industrial revolutions, the convergence of virtual and reality is progressively be- coming a reality. The first and second industrial revolutions brought about innovation in the physical world, whereas the third and fourth industrial revolutions are broadening the area of innovation by integrating the virtual world. Despite the market’s relevance and interest, it is challenging to locate studies that examine the metaverse ecosystem’s com- position. The idea of the metaverse is still developing, and various individuals are adding to its significance in unique ways [22]. As a result, there are numerous definitions of the term ”metaverse” in the majority of earlier studies, and there are currently no established guidelines. It was described as a virtual world by some researchers [16], while others characterised it as the fusion of virtual and actual worlds. Additionally, it appears that not many articles have gone a step farther and accurately researched the ecosystem of the metaverse. Even though J. Y. Kim [17] studied the metaverse ecology his research was only focused on the content ecosystem. Research is therefore restricted to a certain region of the metaverse. The truth is that the metaverse may be difficult to grasp and apply practically if there isn’t an established standard for the ecosystem’s makeup. The metaverse was tackled and studied from the device side in certain studies [16]. There have been additional studies done from the viewpoint of games or contents, but it is challenging to locate one that looked at the metaverse ecology as a whole. The requirement for an approach from the viewpoint of the metaverse ecology was highlighted by W. H. Seok in the year [32]. A compound phrase formed from the words ”meta,” which means virtual and transcendental, and ”universe,” which means world and universe, describes this metaverse or expanded virtual world [13]. ASF characterised the metaverse as ”a universe in which virtual and real constantly interact and co-evolve, in which society and economics form and cultural activities create value” fifteen years later, in 2007. Despite receiving a lot of interest, Lind Lab in the US launched the Second Life metaverse service in 2003. Additionally, when Roblox launched its metaverse service in the US in 2006, 55 percent of American youths under the age of 16 signed up, and by 2021, there were 40 million daily users on average. More than three times as many as Facebook, Instagram, YouTube, and TikTok combined. It turns out to take quite a bit of time [7]. The participant pay system was in the past a weak point in the metaverse economy system. It featured a straightforward revenue model that included social media ads and games. It merely concentrated on transferring the physical environment into a virtual one. In contrast, current and prospective metaverses (such as Roblox’s ”Robux,” Portlight’s ”V-bucks,” ZEPETO’s ”Coin” and ”ZEM,” and Decentraland’s ”MANA”) pay and honour players using its own currency [18]. It is a setting where the platform’s revenue (through subscription models, content/item sales, revenue from adverts, and in-app payment systems) can be used in the actual economy. When creating content, it collaborates with Gucci and Nike to generate B2B revenue. As the four world views are appropriately merged and fused/composited to speed up the interaction, it is usual in practical application to develop a new business model [31]. For instance, Users can wear AR glasses to create a running avatar with the new Ghost Pacer, which was just announced on Kickstarter [23]. The virtual avatar runner sprints in front of the user at a preset speed or at a speed appropriate for his or her previous running. It moves and serves as a pacemaker for the eyes. This is an illustration of how augmented reality (AR) and life journaling have come together. In 2016, Niantic, a Google subsidiary and co-developer of Pokemon GO, conducted research and created related technologies for augmented reality (AR) and virtual reality (VR) [28]. The groundwork has been set to maximise immersion through qualitative advancements in both culture and technology. According to H. S. Ning et al. [27], 96 investigations of the development stage (2020–2021) in the metaverse were conducted from the viewpoints of multiple technologies or ecosystems. The most recent research trend appears to have started looking at the environment of the metaverse. When it comes to the architecture of the metaverse, 2G, 3G, and 4G-based networks were previously supported, to the point where 2D graphics could be sent via mobile phones and a mouse or keyboard. The current and future metaverse, however, offers a 5G communication network environment an NFT that can safeguard the ownership of digital assets,

Comment 2: The authors are suggested to improve the authority of references, and you should cite all papers you use properly.  

Reply: We are thankful for this valuable comment. The changes has been incorporated in the revised manuscript.

Comment 3: This manuscript should be added with the part of Research Limitations.

Reply: Authors thanks the reviewer for the comment. The limitation has been added in the revised manuscript.

Research Limitations

Cyber-physical attacks are one of the major risks to the Metaverse, among all other security issues. Here, an attacker can use cyberattacks to seize control of physical systems. This can entail seizing control of an individual's avatar in order to hurt or change the surroundings. Finding a jurisdiction and a set of laws that can guarantee the users of the virtual environment are safe and secure will be a real problem. The majority of people globally do not have access to broadband internet, which prevents them from taking advantage of the metaverse's full potential. The drawbacks of the metaverse also highlight the need for cutting-edge technology for communication. A premium VR headset needed to access the metaverse is often out of reach for many individuals.

Comment 4: CONCLUSIONS needs more in it, as it's more of an afterthought.   The authors are suggested to highlight important findings and include afterthought of this work.

Reply: Authors thanks the esteemed reviewer for advice. Conclusion section has been enhanced in the revised manuscript by adding the findings of the work.

Comment 5: The authors are suggested to enrich the contents of the whole manuscript including tables, to make it more convincing.

Reply: We are thankful for this valuable comment. The changes has been incorporated in the revised manuscript.

Comment 6: Please carefully check some small writing problems. For example, missing a "." in line 42.

Reply: Author thanks the reviewer for the comments. The manuscript has been thoroughly revised and checked for typographical errors and all such errors have been rectified.

Editor Comments

I believe that the ZTA Model for the Metaverse is a very interesting idea. The current article is suitable for a conference paper as a short review paper. To be a journal paper, it's necessary to provide a little details of the state of the art with a comprehensive review of related works and feasibility of the proposed model. Additionally the authors should provide relevant descriptions on challenges in line with the paper title.

Reply: Authors thanks the esteemed reviewer for the valuable comment. The manuscript has been improved with adding the sections such as literature survey, Future directions for the challenges, Table etc. which increase the overall quality of the manuscript.

Round 2

Reviewer 1 Report

Authors have addressed all review comments